# Cortical–Subcortical Functional Preservation and Rehabilitation in Neuro-Oncology: *Tractography-MIPS-IONM-TMS* Proof-of-Concept Study

**DOI:** 10.3390/jpm13081278

**Published:** 2023-08-20

**Authors:** Francesca Vitulli, Dimitrios Kalaitzoglou, Christos Soumpasis, Alba Díaz-Baamonde, José David Siado Mosquera, Richard Gullan, Francesco Vergani, Keyoumars Ashkan, Ranjeev Bhangoo, Ana Mirallave-Pescador, Jose Pedro Lavrador

**Affiliations:** 1Department of Neurosurgery, King’s College Hospital NHS Foundation Trust, Denmark Hill, London SE5 9RS, UK; francesca.vitulli@nhs.net (F.V.); christos.soumpasis@nhs.net (C.S.); richardgullan@nhs.net (R.G.); francesco.vergani@nhs.net (F.V.); k.ashkan@nhs.net (K.A.); ranj.bhangoo@nhs.net (R.B.); a.mirallave-pescador@nhs.net (A.M.-P.); josepedro.lavrador@nhs.net (J.P.L.); 2Department of Neurosciences and Reproductive and Dental Sciences, Division of Neurosurgery, University of Naples, “Federico II”, Via S. Pansini, 80131 Naples, Italy; 3Department of Neurophysiology, King’s College Hospital NHS Foundation Trust, Denmark Hill, London SE5 9RS, UK; a.diazbaamonde@nhs.net (A.D.-B.); josedavidsiado@gmail.com (J.D.S.M.)

**Keywords:** preoperative mapping, minimally invasive parafascicular surgery, intraoperative neurophysiological monitoring, transcranial magnetic stimulation, rehabilitation, overall survival, quality of life

## Abstract

Surgical management of deep-seated brain tumors requires precise functional navigation and minimally invasive surgery. Preoperative mapping using navigated transcranial magnetic stimulation (nTMS), intraoperative neurophysiological monitoring (IONM), and minimally invasive parafascicular surgery (MIPS) act together in a functional-sparing approach. nTMS also provides a rehabilitation tool to maximize functional recovery. This is a single-center retrospective proof-of-concept cohort study between January 2022 and June 2023 of patients admitted for surgery with motor eloquent deep-seated brain tumors. The study enrolled seven adult patients, five females and two males, with a mean age of 56.28 years old. The lesions were located in the cingulate gyrus (three patients), the central core (two patients), and the basal ganglia (two patients). All patients had preoperative motor deficits. The most common histological diagnosis was metastasis (five patients). The MIPS approach to the mid-cingulate lesions involved a trajectory through the fronto-aslant tract (FAT) and the fronto-striatal tract (FST). No positive nTMS motor responses were resected as part of the outer corridor for MIPS. Direct cortical stimulation produced stable motor-evoked potentials during the surgeries with no warning signs. Gross total resection (GTR) was achieved in three patients and near-total resection (NTR) in four patients. Post-operatively, all patients had a deterioration of motor function with no ischemia in the postoperative imaging (cavity-to-CST distance 0–4 mm). After nTMS with low-frequency stimulation in the contralateral motor cortex, six patients recovered to their preoperative functional status and one patient improved to a better functional condition. A combined Tractography-MIPS-IONM-TMS approach provides a successful functional-sparing approach to deep-seated motor eloquent tumors and a rehabilitation framework for functional recovery after surgery.

## 1. Introduction

Surgical treatment of deep-seated brain tumors is particularly challenging for neurosurgeons and patients because of the dilemma between the oncological advantage of maximal safe resection [1,2,3,4,5,6,7] and the increasing risk of new neurological impairments [8]. 

Resection of highly motor eloquent brain tumors without cortical expression requires precise functional navigation and minimally disruptive surgical approaches. Preoperative mapping using navigated transcranial magnetic stimulation (nTMS), tractography, intraoperative neurophysiological monitoring (IONM), and minimally invasive parafascicular surgery (MIPS) are combined to provide a functional-sparing approach.

Although advances in technology have made brain surgery safer and more effective, patients often experience surgery-related side effects and complications such as cognitive deficits and impaired motor function [9]. These deficits can have a significant impact on the patient’s quality of life (QoL), including their ability to work. 

It is well understood and proven that subcortical injury is the primary cause of neurological deficits in awake craniotomies. Trihn et al. [10] have demonstrated that 90% of new intraoperative neurological deficits occurred during subcortical dissection, against only 2.5% that occurred during cortex manipulation. Furthermore, 43% of these cases experienced worsened deficits in the immediate postop period and 14% continued to have worsened deficits at 3-month follow-up. Therefore, sparing subcortical areas during resection may reduce the severity of both immediate and late neurological deficits. This is the designed purpose of the MIPS techniques, aiming to preserve the subcortical white matter tracts. As described by Kassam et al. [11], the access is always trans-sulcal, using the natural corridors for accessing the targeted lesions, thus minimizing the disruption of the healthy cortical tissue. The trajectory is parafascicular, running parallel to the white matter tracts, eliminating all the shear forces. 

Neurorehabilitation aims to restore these lost functions through targeted exercises and techniques. One promising technique that has emerged in recent years is transcranial magnetic stimulation (TMS). It provides a rehabilitation tool that can potentially maximize functional recovery after functional-eloquent surgery [12].

The aim of this article is to present the results of a proof-of-concept study that employed an integrative approach for cortical–subcortical functional preservation and rehabilitation in neuro-oncology.

## 2. Materials and Methods 

This is a single-center prospective proof-of-concept cohort study between January 2022 and June 2023 of patients admitted for surgery with motor eloquent deep-seated brain tumors. 

Our inclusion criteria were the following: adult patients admitted for highly motor eloquent subcortical tumor surgery—invasion/contact with corticospinal tract (CST) documented with preoperative lesion-to-CST distance < 1 mm, nTMS preoperative mapping, and tractography; minimal invasive parafascicular surgery (MIPS); intraoperative neurophysiological monitoring; and nTMS after surgery for motor rehabilitation. Patients of any age meeting all criteria were included. The exclusion criteria were incomplete data. Data collected for each patient were age at diagnosis, clinical and radiological features, surgical approach, extent of resection, histological diagnosis, rehabilitation, and outcomes. 

Figure 1 summarizes the multistage approach used in this study.

Preoperative nTMS was performed for the identification and characterization of the primary motor cortex. This information was integrated into the surgical planning to establish a safe outer corridor for the MIPS approach [13].

The MIPS procedure consisted of a transulcal approach using either the superior frontal or the intraparietal sulci. NICO BrainPath© System and NICO Myriad© microdebrider were used for tumor resection. Electrification of the microdebrider was performed as per a previously published technique [14]. 

Tractography was performed with a constrained spherical deconvolution algorithm using Cranial Medtronic©. Region-of-interest-based dissection of the corticospinal tract (CST)—precentral gyrus and ipsilateral mesenchephalon; fronto-striatal tract (FST)—supplementary motor area (SMA) and caudate nucleus; fronto-aslant tract (FAT)—pre-SMA and inferior fronto-operculum (*pars opercularis*); and *cingulum*—frontal cingulate and parietal cingulate subcortical area, was performed. Pre-processing using the Cranial application of STEALTH Medtronic was performed using eddy current correction. For the purpose of this study, lesion segmentation was performed. The T1-weighted Gadolinium MRI volumetric scan was used for the segmentation, as all lesions were contrast-enhancing. Co-registration of preoperative and postoperative structural imaging (MRI T1 Gad) was performed for the calculation of the shortest distance between the tumor or the cavity and the tract (lesion-to-tract and cavity-to-tract distances). 

Our standard surgical protocol was used in all cases: alcoholic betadine and chlorhexidine prep and sterile drape; skin covered by Ioban; local anesthetic with adrenaline instilled; craniotomy planned with STEALTH. The dura was incised according to the tumor site and specific anatomical features. Cortical mapping was performed with the monopolar probe over the exposed cortex. An electrode strip was inserted over the presumed position of M1 and the correct placement was confirmed with positive responses. ICG was performed. Sulcal splitting was performed with the scissors and the bipolar. The tubular retractor system (size according to the depth of the lesion) was inserted in the split sulcus and secured on the BUDDE halo. The correct position was confirmed with STEALTH (Figure 2).

The MIPS approach in the 4 patients with the mid-cingulate lesions involved a trajectory through the fronto-aslant tract (FAT) and fronto-striatal tract (FST). Continuous neuro-monitoring and direct cortical stimulation were utilized to evaluate the motor-evoked potentials. Progressive debulking of the tumor with bipolar, suction, and NICO Myriad© microdebrider was performed. Hemostasis of the surgical cavity was achieved prior to the retraction of the tube. ICG was performed after the removal of the tubular retractor to identify the arterial and venous perfusion. 

We performed a multimodal intraoperative neuromonitoring (IONM) using direct cortical stimulation for motor-evoked potentials (MEPs) and dynamic subcortical mapping with a suction probe (spesmedica reg) or a modified microdebrider that was electrified with a sterile crocodile clip for subcortical identification of the corticospinal tract. The same strip electrode that was used to perform direct cortical stimulation (DCS) was used to record cortical somatosensory evoked potentials (SSEPs) and electrocorticography (ECoG). Transcranial MEPs and SSEPs were also utilized to record opening baselines and to assess for ischemia at the end of the resection. Free-running EMG was recorded to confirm the electro-clinical correlation of possible seizures in case there were changes in the electrocorticography (EEG).

Postoperative imaging was acquired within 72 h of the surgical procedure. Diffusion-weighted images were acquired on a 3-T MRI scanner using a cardiac-gated single-shot spin-echo echo-planar imaging multiband sequence (TE 80 msec, TR 4000 msec) along 90 diffusion directions with a b-value of 2500 sec/mm^2^ (FOV 256 × 256 mm^2^).

Navigated repetitive TMS (nrTMS) was performed as per a previously published protocol by Ille et al. [12]. Low-frequency stimulation was applied to the contralateral hotspot for the upper limb in a total of 900 stimulations at 1 Hz for 15 min at 100% of the resting motor threshold for 7 consecutive days. In all patients, the nTMS hotspot was located in the precentral gyrus.

Written informed consent was obtained from all the parents of the patients for preoperative and intraoperative mapping procedures as well as postoperative nTMS for motor rehabilitation. The manuscript was conducted ethically in accordance with the World Medical Association Declaration of Helsinki.

### Statistical Analysis

The raw data were entered into Microsoft Excel (Version 10.14 for Mac). Statistical analyses were performed via R (version 4.0.2; The R Foundation for Statistical Computing) and RStudio (version 1.2.1335). Standard descriptive statistics were used to describe the characteristics of cases (median with range, mean ± SD, and frequencies with percentages). 

## 3. Results

The study enrolled seven adult patients, five females, and two males. The mean age was 56.3 years old (range: 26–79 years old). Four patients presented with motor weakness, one of them with expressive dysphasia, one with simple focal seizures, and one only with headaches. On clinical examination before surgery, all patients experienced motor deficits (ranging from 0 to 4/5 MRC hemiplegia/hemiparesis).

In three patients the lesion was located at the cingulate gyrus. In two of them, the tumor was centered in the precentral gyrus, and in the final two, at the basal ganglia involving the thalamus and the internal capsule. Four patients had left-sided lesions and three had right-sided lesions. The mean distance between the lesions and the cortex as measured in the preoperative MRIs was 30.9 mm (range: 7 mm–46 mm) (Figure 3 and Figure 4). The mean volume of the included lesions was 13.2 cm^3^ (range: 3.3–32.6 cm^3^).

The most common histological diagnosis was metastasis in five patients; one patient had a pilocytic astrocytoma (Grade 1 WHO) and one GBM (IDH wild type, ATRX preserved, MGMT 9%, Grade 4 (WHO)). 

Intra-operatively, the mean distance from CST after subcortical stimulation with the monopolar probe was 3.6 mA (range: 2–5 mA). In all cases, there was no deterioration of the MEPs during the operation. In two cases, there was an improvement in the amplitude of the potentials after the resection of the lesion. 

Gross total resection (GTR) was achieved in three patients, and near-total resection (NTR) in four patients (Figure 5). No ischemia was shown in the postoperative imaging and the range of cavity-to-CST distance was 0–4 mm. 

Post-operatively, in all patients the motor power in the affected side deteriorated. More specifically, two patients had a drop of three grades (from 4/5 to 1/5 and from 3/5 to 0) and one patient had a drop of one grade (from 4/5 to 3/5) in the MRC scale in both upper and lower limbs. One patient had a drop of three grades only in the upper limb (from 3 to 0) and one deteriorated by two grades only in the lower limb (from 2/5 to 0/5). One patient had a postoperative deficit affecting only the fascial muscles and one patient remained unchanged. The immediate postoperative deterioration of motor function was attributed to initiation (trans-FAT or FST approach for mid-cingulate lesions) and manipulation close to CST (central core lesions). 

In the cohort of the patients with metastatic brain disease, after navigated repetitive TMS with low-frequency stimulation in the contralateral motor cortex, four patients recovered to their preoperative functional status—performance status (PS) 0 or 1 and one patient improved to a better functional condition—PS 0 from PS 1 (Figure 6). The patient with the pilocytic astrocytoma and the GBM recovered at the preoperative performance status—PS 0 and 1, respectively. One of the patients with a histological diagnosis of brain metastasis, who was hemiplegic pre- and immediately post-operatively, has now recovered to motor power 3/5 in both upper and lower limbs after 10 days of nTMS.

The mean length of hospital stay for the other six patients was 18.5 days (range: 3–56 days). Two of them were discharged to a rehabilitation unit and the other four were able to return home. It is important to highlight that the reason for the delayed discharge of the patient with the longer hospital stay (56 days) was a lack of available beds in the rehabilitation unit. Furthermore, in two patients the hospital stay was deliberately prolonged in order to continue the postoperative TMS-assisted rehabilitation.

All the clinical and peri-operative details of the seven patients can be found in Table 1.

## 4. Discussion 

The ultimate goal in neuro-oncology resection surgery is to achieve the maximal safe tumor resection in order to accomplish a prolonged survival with simultaneous preservation of the neurological function. The above becomes increasingly challenging in deep-seated highly eloquent tumor cases in close proximity to the white tract motor pathways. Our integrated approach combines treatment planning at three consecutive levels: preoperative—nTMS and tractography; intraoperative—navigation and intraoperative neuromonitoring and mapping; and postoperative—nrTMS for rehabilitation. Together, these technological advances support an efficacious functional-sparing technique to facilitate maximal surgical resection. 

Our protocol used the framework published by Jennings et al. [13] and goes one step further with the introduction of the nTMS for rehabilitation after the surgical procedure. This strategy incorporates a combination of nTMS for pre-surgical identification and characterization of the primary motor cortex (outer corridor) with tractography of motor eloquent projection—CST and FST—and association—FAT—tracts (inner corridor) to define the surgical approach and the location of the craniotomy. Furthermore, it can define the correlation of the lesion with the eloquence of the surrounding brain, thus establishing safe resection margins.

The impact of the above-mentioned techniques on motor outcomes has been previously established. Krieg et al. [15] showed that preoperative TMS can lead to better functional outcomes with a statistically significantly lower number of neurologic deficits in patients where TMS was utilized. The study from Sollmann et al. [16] was one of the first to demonstrate a statistically significant correlation between postoperative motor deficit and the lesion-to-CST distance when a nTMS-based tractography technique was used. In our study, the lesion-to-CST distance, based on the preoperative imaging protocol used, was <1 mm for all the included patients, thus predicting a significant risk for a postoperative deficit and motor score deterioration. 

Intraoperative neurophysiological monitoring and mapping is currently an irreplaceable tool in oncology surgery. The utilization of intraoperative monitoring can facilitate both the maximization of the resection margins and the reduction of postoperative neurologic deficits [17]. This proof-of-concept study supports that the combined use of nTMS, tractography, and IONM is effective in the anatomical preservation of the motor pathway given the avoidance of a trajectory through the primary motor cortex and the avoidance of transgression of the CST during the resection due to the intraoperative use of navigation and IONM. The latest was achieved in our study with continuous subcortical stimulation with the monopolar probe. In our cases, the distance between the surgical margins and the CST varied between 2 and 5 mA, with a mean of 3.6 mA. 

The anatomical preservation of the structural networks that sub-serve the motor function is crucial for potential motor recovery and rehabilitation after surgery. Multiple authors [18,19,20] have demonstrated that MEP deterioration (direct cortical stimulation from a strip electrode) during surgery can be consistently correlated with postoperative motor deficits, establishing MEP monitoring as the most reliable method of intraoperative motor function surveillance in glioma surgery under general anesthesia. As per the study by Zhou et al. [21], a reduction of 50% or more in MEPs is associated with postoperative deficits. However, more than half of the patients can have motor deterioration with preserved MEPs at the end of surgical resection [20,22]. The latest was the pattern seen in the patients included in our study. Although MEPs remained stable intra-operatively, all of them developed motor deficits in the affected side post-operatively. Different factors can be responsible for that, such as secondary hemorrhage, ischemia, or resection of the supplementary motor area [18].

Multiple studies have shown the potential of nTMS in monitoring and predicting functional recovery after functional-eloquent brain surgery [9,23]. Ivren et al. [24] found that nTMS was able to predict the recovery of motor function in patients who underwent surgery for high-grade gliomas associated with the motor cortex (M1) or the corticospinal tract (CST). The study compared the results of nTMS with the results of clinical tests and found that nTMS was more sensitive in predicting the recovery of motor function [24].

Moreover, TMS is proven to facilitate neuroplasticity, the brain’s ability to reorganize and form new neural connections in response to changes in its environment, therefore being an attractive tool in neurorehabilitation after stroke for the treatment of both acute and chronic motor and language deficits with positive results [25,26,27,28,29,30]. 

In recent years, several studies have investigated the use of TMS in postoperative neurorehabilitation following brain surgery [12,31,32,33,34]. Ille et al. [12] performed a randomized, double-blinded trial to examine the effect of TMS therapy in 22 patients suffering from acute surgery-related functional deficits after glioma resection. The authors concluded that TMS has significantly contributed to the improvement of motor impairment and general oncological and comprehensive neurological outcome parameters; in addition, the procedure was deemed safe and was well-tolerated by all patients with no adverse effects such as seizures occurring during or after TMS. All the above studies support our decision to implement nTMS as part of our surgical protocol in highly eloquent motor-related lesions. 

Our study used nTMS for motor rehabilitation after successful preservation of the motor pathway using a preoperative and intraoperative functional-informed MIPS technique. This was confirmed by positive nTMS responses in the postoperative mapping for every patient included and by the motor recovery experienced by these patients with the nrTMS for rehabilitation. As rightly pointed out by Schmidt et al. [33], we believe the assumption “surgery can do no harm” might be replaced with the motto “perioperative stimulation can facilitate early recovery”. The fact that every patient recovered to their baseline or better functional status after surgery for a deep-seated tumor involving the CST with an NTR-GTR in six out of seven patients supports the onco-functional balance provided by this integrated surgical technique. Additionally, by extending the surgical indication [35,36,37], this technique provides surgical treatment to patients that would be previously excluded due to unsatisfactory planning, operative, or rehabilitation techniques.

### Limitations and Strengths

Our study has not been able to fully incorporate other preoperative imaging modalities such as fMRI and magnetoencephalography (MEG). fMRI is a validated tool for preoperative planning in neuro-oncology patients. Many studies in the literature up to now have demonstrated that it can be very sensitive in detecting functional motor cortex areas adjacent to a focal brain lesion [38,39]. On the other hand, there is existing literature that supports that nTMS as a preoperative mapping tool has a higher spatial resolution for the motor areas compared to fMRI. For example, Coburger et al. [40] demonstrated that nTMS was more accurate to localize the motor cortex compared to fMRI. This was more evident in the subgroup of intrinsic tumors, where the lower extremity localization had significantly higher accuracy scores when nTMS was used compared to the use of fMRI. This is the main reason why in our department we prefer the use of nTMS for motor mapping. Another advanced technique that could play a more important role in the future is MEG, which shows some promising results as part of pre-surgical planning in eloquent brain lesions [41]. Furthermore, another crucial tool for better neurological outcomes in oncology patients is awake surgery [42] with intraoperative monitoring of the patient, the impact of which has not been assessed in this series. 

Another limitation of our study is the heterogeneity of the histological diagnoses in our patients. However, the main focus of this study is the functional outcome and not the overall survival. It is true that in a small cohort, survival outcomes cannot be measured due to the heterogeneity of the patients included. Nevertheless, surrogate information can be suggested based on the impact of the extent of resection in the included pathologies as these were lesions that potentially could not be resected or would not have been resected.

Additionally, it is important to highlight that there is uncertainty regarding the impact of nTMS in rehabilitation as our patients were not randomized to this intervention. However, the previous randomized trial by Ille et al. [12] supported the treatment effect of nTMS in terms of motor deficit improvement in the patients included. 

Finally, our study is limited by the small number of patients; thus we are not able to conclude any statistically significant results. It is important to notice that the aim of this work is to provide a proof-of-concept research idea and evaluate the feasibility of this peri-operative framework, without comparing different peri-operative surgical strategies. Furthermore, we should notice that highly motor eloquent tumors patients that proceed with surgery are few and this is the main reason why a proof-of-concept study was performed. 

Further work is required to assess differences in the impact of this strategy in terms of the different histological diagnoses of the tumors as well as the specific tumor grading. This could be achieved by adopting a multicenter setting given the low number of patients assessed by a single center of patients with such highly eloquent lesions.

Nevertheless, and as far as we are aware, this is the first study in the literature incorporating an extensive and detailed surgical protocol based on pre-, intra-, and postoperative stages with enhanced TMS-based rehabilitation. 

## 5. Conclusions

This study demonstrated that an integrated approach involving a combined Tractography-MIPS-IONM-TMS is a valuable tool for preserving cortical–subcortical functional anatomy during surgery, minimizing operative morbidity, and maximizing postoperative functional outcomes in patients with deep-seated motor eloquent tumors. The protocol is a safe and feasible technique that can be integrated into everyday practice, allowing patients to benefit from a successful functional-sparing approach and a rehabilitation framework for functional recovery after surgery. The results of this proof-of-concept study are promising and warrant further investigation in future research.

## Figures and Tables

**Figure 1 jpm-13-01278-f001:**
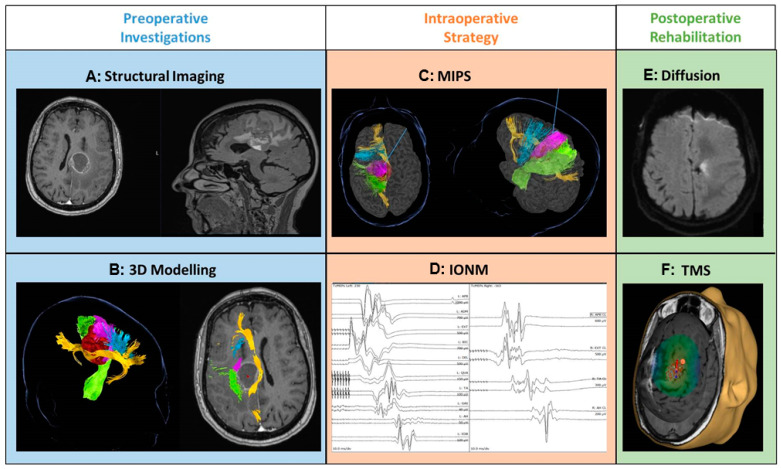
Tractography-MIPS-IONM-TMS proof-of-concept study framework: Left: preoperative investigations: (**A**) structural imaging (T1 gad axial and sagittal MRI brain); (**B**) preoperative tractography (green: corticospinal tract, blue: fronto-aslant tract, purple: fronto-striatal tract, yellow: cingulum, red: tumor). Middle: intraoperative strategy: (**C**) functional corridors for the minimal invasive parafascicular approach (MIPS); (**D**) intraoperative neuromonitoring and mapping (IONM). Right: postoperative approach and rehabilitation: (**E**) postoperative diffusion imaging (DWI axial MRI brain); (**F**) transcranial magnetic stimulation mapping for motor assessment and rehabilitation (low-frequency stimulation applied to the contralateral hotspot).

**Figure 2 jpm-13-01278-f002:**
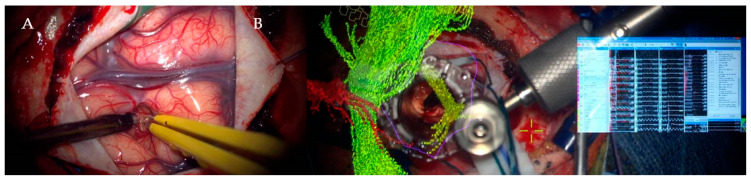
Intraoperative minimal invasive parafascicular surgery (MIPS) procedure: (**A**) Sulcal dissection prior to cannulation of the brain. (**B**) Integrated visualization of tumors and tractography as well as intraoperative neuromonitoring and mapping during MIPS.

**Figure 3 jpm-13-01278-f003:**
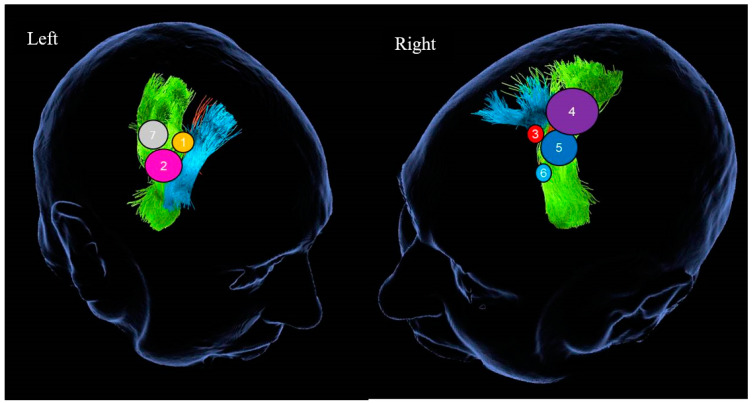
Schematic representation of tumor location. Left image—patients with right-sided lesions. Right image—patients with left-sided lesions. Each number corresponds to the patient’s number in Table 1. The volume of the sphere approximates the volume of the specific lesion. Green—corticospinal tract; blue—fronto-aslant tract; orange—fronto-striatal tract.

**Figure 4 jpm-13-01278-f004:**
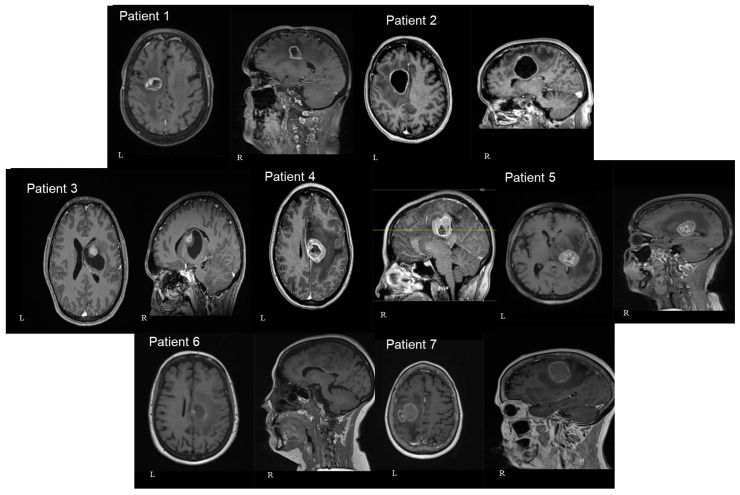
Axial (left—L) and sagittal (right—R) T1 gad MRI brain images of all seven patients included in our study. In 3 patients the lesion was located at the cingulate gyrus (numbers 2, 4, and 6). In 2 of them, the tumor was centered in the precentral gyrus (numbers 1 and 7). In the final 2 patients, the lesion was centered at the basal ganglia involving the thalamus and the internal capsule (numbers 3 and 5). The number of patients in each image corresponds to the patients mentioned in Table 1.

**Figure 5 jpm-13-01278-f005:**
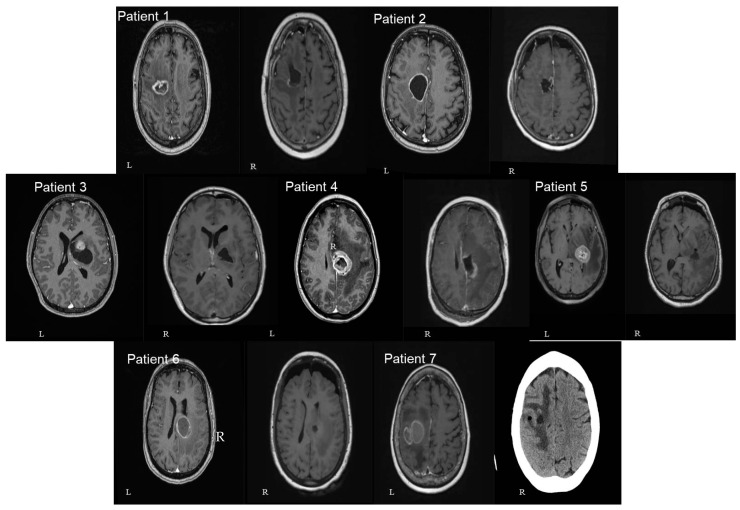
Axial pre (left—L) and postoperative (right—R) T1 gad MRI brain images of all seven patients included in our study. A single axial view was selected in order to avoid confusion with multiple images. The number of patients in each image corresponds to the patients mentioned in Table 1 (patient number 7 had only postoperative CTH and no MRI available).

**Figure 6 jpm-13-01278-f006:**
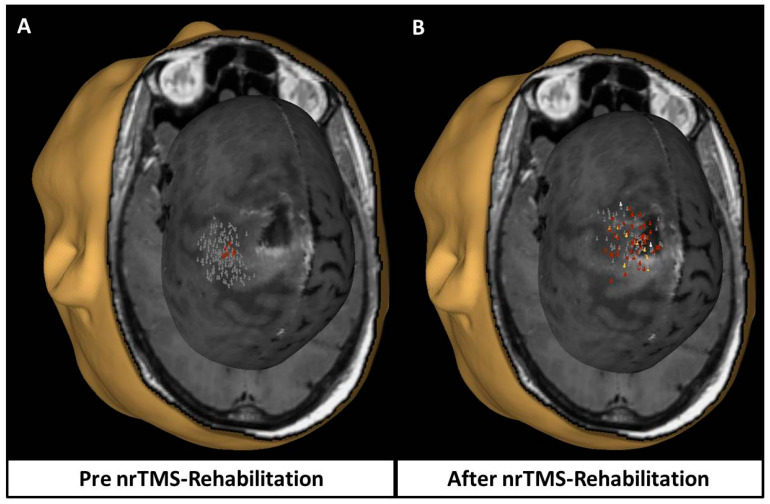
Impact of navigated repetitive transcranial magnetic stimulation in upper limb motor mapping. (**A**) Responses before nrTMS rehabilitation. (**B**) Responses after nrTMS rehabilitation. Significant increase in the cortical representation of the motor area for the upper limb with an increase in the amplitude of the elicited motor-evoked potentials after 7 days of treatment with low-frequency stimulation. Grey—negative responses; red—positive responses >50 microV and < 500 microV; yellow—positive responses > 500 microV and < 1000 microV; white—positive responses > 1000 microV.

**Table 1 jpm-13-01278-t001:** F: female; M: male; PS: performance status; CST: corticospinal tract; EOR: extent of resection; TMS: transcranial magnetic stimulation; LUL: left upper limb; LLL: left lower limb; RUL: right upper limb; RLL: right lower limb; GTR: gross total resection; NTR near-total resection.

Patient	Age, Sex	Presentation		Volume (cm^3^)	Preop Motor Deficit	Preop PS	Tumor-to-CST Distance	EOR	Histological Diagnosis	Postop Motor Deficit	Post TMS Motor Deficit	Follow-Up PS	Follow-Up (Months)	Status
Patient 1	56, F	Simple focal seizure involving the head, face, and upper limb	Right precentral gyrus	6.24	LUL 3/5, LLL 4/5	1	<1 mm	NTR	Metastatic carcinoma, lung primary (high Ki67, PDL1 positive)	LUL 0/5, LLL 4/5	LUL 5/5, LLL 5/5	2	8	Alive
Patient 2	60, F	Left upper limb weakness, forgetfulness/brain fog for a few months	Mid-cingulate gyrus	19.1	LUL 4/5, LLL 4/5	1	<1 mm	NTR	GBM, IDH wild type, ATRX preserved, MGMT 9%. Grade 4 (WHO)	LUL 3/5, LLL 3/5	LUL 5/5, LLL 5/5	1	1	Alive
Patient 3	26, M	Headaches	Anterior limb of the internal capsule	3.51	None	0	<1 mm	GTR	Pilocytic astrocytoma, Grade 1 WHO (Ki67 1%)	R facial palsy	RUL 4/5, RLL 4/5	0	9	Alive
Patient 4	39, F	Right-side hemiparesis and headaches	Mid-cingulate girus	32.6	RUL 3/5, RLL 2/5	1	<1mm	NTR	Metastatic Malignant Melanoma (Ki67 15%)	RUL 3/5, RLL 0/5	RUL 4/5, RLL 3/5	1	3	Died
Patient 5	67, M	Expressive dys/aphasia, wordfinding difficulties	Posterior limb of the internal capsule	15.9	RUL 4/5, RLL 4/5	1	<1 mm	GTR	Metastatic adenocarcinoma, breast primary (CK7 positive, CK20 negative, CDX2 negative, TTF1 negative, GCDFP15 positive)	RUL 1/5, RLL 1/5	RUL 4/5, RLL 4/5	0	3	Alive
Patient 6	79, F	Right side weakness, impaired mobility	Mid-cingulate gyrus	3.3	RUL 3/5, RLL 3/5	1	<1 mm	NTR	Metastatic malignant melanoma (Ki67 40%, PDL1 negative)	RUL 0/5, RLL 0/5, aphasia	RUL 4/5, RLL 4/5	1	4	Died
Patient 7	67, F	Left upper and lower limb weakness	Right precentral gyrus	11.8	LUL 0/5, LLL 0/5	1	< 1mm	GTR	Metastatic malignant melanoma (BRAF +)	LUL 0/5, LLL 0/5	LUL 3/5, LLL 3/5	1	Inpatient	Alive

## Data Availability

Data are available upon request.

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
