# Peer review of "Cortical–Subcortical Functional Preservation and Rehabilitation in Neuro-Oncology: Tractography-MIPS-IONM-TMS Proof-of-Concept Study"

_jpm, 2023, doi:10.3390/jpm13081278_

Round 1

Reviewer 1 Report

This manuscript is well written and topic of interest for neurosurgeon.

Author Response

Thank you very much for you review.

Reviewer 2 Report

This manuscript provides an integrated combination of diffusion MRI-based tractography, minimally invasive parafascicular surgery, operative monitoring, and nTMS for brain tumor surgery for patients.  This study could provide critical values for the rehabilitation strategy for future brain surgery. It is well written. However, some concerns need to be further addressed:

Major comments:

1.  Please clarify did you perform lesion or tumor tissue segmentation. If so could elaborate a bit more on that, and how did you define the boundary of the tumors and the distance between the CST and the tumor?

2 . Besides the distance to the CST, I think the lesion or tumor size is also an important factor, please add the tumor size in Table 1. And add some discussion about that.

3. Did you acquire the T2FLAIR images for patients? For the details of the MRI protocol, please add the parameters of the structural and diffusion MRI as well as the details for DWI image processing and fiber tract reconstruction.

4. In Table 1. the distance between Tumor and CST was all 1 mm, but are there cases with an invasion of the tumor to the CST?  

4. For the nTMS hotspot, please add some more details of the location, is it the same for every subjects?

5. For Figure 3, the location of the tumor. is that just the mass center of the tumor? Please modify it to a tumor distribution map or probability map. 

Minor:

1. In the introduction part,  The author claims that " It provides a rehabilitation tool to maximize functional recovery after functional-eloquent surgery [12]", This statement is subject to debate.

2. Please add table legend for Table 1.

3.  Fig4, the contrast of image is too low

4. Please add Left and Right label for figure 3 and figure 4

Author Response

Thank you very much for tour review.

       1. Please clarify did you perform lesion or tumor tissue segmentation. If so could elaborate a bit more on that, and how did you define the boundary of the tumors and the distance between the CST and the tumor?

Authors: Thank you very much for your comment. We performed lesion segmentation. We used the T1 Gad volumetric scan for segmentation, as all lesions were contrast enhancing. We have calculated the shortest distance between the lesion and the CST and assumed that value. The above comments have been added in the methods of our manuscript.

  1. Besides the distance to the CST, I think the lesion or tumor size is also an important factor, please add the tumor size in Table 1. And add some discussion about that.

Authors: Thank you very much for your comment. The volumes of all the lesions are now included in the table 1 and a discussion about that has been added in the results sections of our manuscript.

  1. Did you acquire the T2 FLAIR images for patients? For the details of the MRI protocol, please add the parameters of the structural and diffusion MRI as well as the details for DWI image processing and fiber tract reconstruction.

Authors: Thank you very much for your comment. Unfortunately, we don’t have available volumetric pre-operative T2 FLAIR for all patients and therefore information based on this sequence was not included.

Diffusion-weighted images were acquired on a 3-T MRI scanner using a cardiac-gated single-shot spin echo echo-planar imaging multiband sequence (TE 80 msec, TR 4000 msec) along 90 diffusion directions with a b-value of 2500 sec/mm2 (FOV 256 × 256 mm). 

Regarding the tractography processing, pre-processing using the Cranial application of STEALTH Medtronic was performed using eddy current correction.

Finally, all the information regarding the ROIs used are already in the method section of our manuscript.

  1. In Table 1, the distance between Tumor and CST was all 1 mm, but are there cases with an invasion of the tumor to the CST?  

Authors: Thank you very much for your comment. As described in table 1, the distance between the lesion and the CST was <1mm for all the cases. Therefore, there was contact/invasion in all patients.

  1. For the nTMS hotspot, please add some more details of the location; is it the same for every subjects?

Authors: Thank you very much for your comment. In all patients, the nTMS hotspot was located in the pre-central gyrus.

  1. For Figure 3, the location of the tumor is that just the mass center of the tumor? Please modify it to a tumor distribution map or probability map. 

Authors: Thank you very much for your comment. Based on your suggestion, we have altered this figure by replacing the spheres, so as each of them to correlate with the volume of each separate lesion.

Minor:

  1. In the introduction part, The author claims that " It provides a rehabilitation tool to maximize functional recovery after functional-eloquent surgery [12]", This statement is subject to debate.

Authors: Thank you very much for your comment. The above sentence has been modified in the manuscript.

  1. Please add table legend for Table 1.

Authors: Thank you very much for your comment. An appropriate legend has been added to the table.

  1. Fig4, the contrast of image is too low

Authors: Thank you very much for your comment. Unfortunately, this is the best possible image that can be generated from the TMS software.

  1. Please add Left and Right label for figure 3 and figure 4

Authors: Thank you very much for your comment. Left and right labels were added for the above images accordingly.

Reviewer 3 Report

Thank you for this manuscript. You present 7 patients that have been thoroughly preoperatively investigated by intraoperative MIPS technique. They have also been preoperatively investigated by nTMS to localize the eloquent cortex. You state, perfectly correctly that the white mater tracts give the most serious neurological deficits when interfered with.

I have some important comments.

First, I think that your approach of mixing pre-, per- and postoperative TMS gives, at least me, some confusion regarding what your message really is? When you plan a neurosurgical procedure in an eloquent area nowadays, it should be routine to use the preoperative mapping techniques available. (In that respect it is noteworthy that you have not included fMRI which has been in clinical use for so many years and gives at least good surface information).

Furthermore, in my opinion, with regard to your Figure 3, that this gives too little information on the tumours for the reader to be able to evaluate the usability of your concept. You have only 7 patients, and with such a small number you have the possibility to show at least the most important parts of each patient’s MRI-scan.

You have patients with very different histopathological diagnosis and no information about the patients with metastasis. This is of course very important information since you also stress the outcome. According to what type of metastasis the five patients had, the expected clinical course would clearly be very different. Then you have one of the most benign tumours, a pilocystic astrocytoma (Grade I WHO), and one of the most malignant tumours GBM IDH wild type (Grade IV WHO).

If you would merely use your method to demonstrate the help that your strategy gives for the surgical procedure, the different etiologies do not make any difference, if you would focus on the possibility of getting an optimal resection without giving the patient a new neurological deficit. In addition you should also, in this small patient cohort, add illustrations of the postoperative MRIs.

In your description of the postoperative course, to me it would be mandatory to divide them according to the histopathological diagnosis. The expected postoperative course for a patient with one of the most benign tumours is of course completely different from the postoperative course of a patient with the most malignant type of brain rumour (GBM), where a recurrence that could start very early would affect the evaluation of the postoperative situation.

Regarding the use of nTMS in the postoperative phase, you give some references to important work on this, but it is not possible to know whether nTMS helped your patients or not.

So, in my opinion, to improve the validity of your results you should shift strategy. The most preferable change would be to increase the number of patients and choose a more homogenous group regarding the histopathological diagnosis. Another approach would be to  focus on the result of the surgery only, without inclusion of the long-term outcome, and then describe the result of the extent of the resection, and if unexpected new neurological deficits actually occured.

Author Response

Thank you very much for your review.

  • First, I think that your approach of mixing pre-, per- and postoperative TMS gives, at least to me, some confusion regarding what your message really is? When you plan a neurosurgical procedure in an eloquent area nowadays, it should be routine to use the preoperative mapping techniques available. (In that respect it is noteworthy that you have not included fMRI which has been in clinical use for so many years and gives at least good surface information).

Authors: Thank you very much for your comments. We totally understand that the use only of pre-operative nTMS as a mapping tool can cause some confusion. However, there are several studies in the literature that support that nTMS as a pre-operative mapping tool has a higher spatial resolution for the motor areas compared to fMRI. For example, Coburger et al1 demonstrated that nTMS was more accurate to localize motor cortex compare to fMRI. This was more evident in the subgroup of intrinsic tumors, where the lower extremity localisation had significant higher accuracy scores when the nTMS was used compare to the use of fMRI. This is the reason why in our centre, we prefer the use of nTMS for motor mapping. Furthermore, it is well known that contrast enhancing tumours alter the brain blood barrier and the cerebral blood volume and perfusion which may impair the spatial resolution in the vicinity of the tumours. The above mentioned comments have been added to the limitations section of our discussion.

References:

  1. Coburger, J., Musahl, C., Henkes, H., Horvath-Rizea, D., Bittl, M., Weissbach, C., & Hopf, N. (2013). Comparison of navigated transcranial magnetic stimulation and functional magnetic resonance imaging for preoperative mapping in rolandic tumor surgery. Neurosurgical review, 36(1), 65–76. https://doi.org/10.1007/s10143-012-0413-2

  • Furthermore, in my opinion, with regard to your Figure 3, that this gives too little information on the tumours for the reader to be able to evaluate the usability of your concept. You have only 7 patients, and with such a small number you have the possibility to show at least the most important parts of each patient’s MRI-scan.

Authors: Thank you very much for your suggestion. We have now added an extra picture (figure 4) showing axial and sagittal images for all the patients included in the study.

  • You have patients with very different histopathological diagnosis and no information about the patients with metastasis. This is of course very important information since you also stress the outcome. According to what type of metastasis the five patients had, the expected clinical course would clearly be very different. Then you have one of the most benign tumours, a pilocystic astrocytoma (Grade I WHO), and one of the most malignant tumours GBM IDH wild type (Grade IV WHO).

Authors: Thank you very much for your comments. Indeed we have patients with different histopathological diagnosis. However, the main focus of the paper is the functional outcome and not the overall survival. We totally agree that in a small cohort, survival outcomes cannot be measured due to the heterogeneity of the patients included. Nevertheless, surrogate information can be suggested based on the impact of extent of resection in the included pathologies as these were lesions that potentially could not be resected or would not have been resected. All the above mentioned comments have now added to the limitations section of our discussion.

  • If you would merely use your method to demonstrate the help that your strategy gives for the surgical procedure, the different etiologies do not make any difference, if you would focus on the possibility of getting an optimal resection without giving the patient a new neurological deficit. In addition you should also, in this small patient cohort, add illustrations of the postoperative MRIs.

Authors: Thank you very much for your suggestion. A separate image including a pre and postoperative axial T1 gad image for all patients has been included in order to demonstrate the extent of resection, figure number 5.

  • In your description of the postoperative course, to me it would be mandatory to divide them according to the histopathological diagnosis. The expected postoperative course for a patient with one of the most benign tumours is of course completely different from the postoperative course of a patient with the most malignant type of brain rumour (GBM), where a recurrence that could start very early would affect the evaluation of the postoperative situation.

Authors: Thank you very much for your suggestion. We have not divided the outcomes initially because the outcomes we report are at 7 days post op where a significant recurrence is less likely. However, we believe your suggestion is meaningful. Therefore we have now separated the outcomes as per pathology in the results section.

  • Regarding the use of nTMS in the postoperative phase, you give some references to important work on this, but it is not possible to know whether nTMS helped your patients or not. So, in my opinion, to improve the validity of your results you should shift strategy. The most preferable change would be to increase the number of patients and choose a more homogenous group regarding the histopathological diagnosis. Another approach would be to  focus on the result of the surgery only, without inclusion of the long-term outcome, and then describe the result of the extent of the resection, and if unexpected new neurological deficits actually occured.

Authors: Thank you very much for your comments.

Response:

  1. Low frequency TMS is a well-established technique for motor rehabilitation. The recent RCT referenced in the manuscript (Ille et al, 2021, reference number 12) provided insight on the utility of this technique in patient-centred recovery of motor function.
  2. In this study we do not provide long-term outcomes. All the motor outcomes and discharge information is provided 7 days after the treatment was started which happened 12-72 hours after surgery.
  3. This is a proof-of-concept study for an integrated perioperative approach to improve motor outcomes in deep seated eloquent lesions. Our aim was to evaluate the feasibility and impact of this technique in a preliminary cohort. We do agree that the motor function and extent of resection depends as well upon the diagnosis and not only the location. This subgroup analysis should be assessed potentially in a multicentre setting given the low number of patient assessed by a single centre of patients with such high eloquent lesions that present in a performance status that allow and consent for a treatment strategy with potential high risk of morbidity and prolonged recovery (7 days of intensive rehabilitation treatment) as an inpatient.

All the above have been included in the limitations section of our discussion.

Reviewer 4 Report

The manuscript titled "cortical-subcortical functional preservation and rehabilitation in neuro-oncology: tractography-MIPS-IONM-TMS proof of concept study" reports an interesting study on an integrated approach involving a combined Tractography-MIPS-IONM-TMS is a valuable tool for preserving cortical-subcortical functional anatomy during surgery, minimizing operative morbidity, and maximizing postoperative functional outcomes in patients with deep seated motor eloquent tumors. The adopted protocol looks interesting for postoperative functional outcome, however, significance of the data is biggest concern.

1.     Figure 1. The images are blurry. The intraoperative neuromonitoring and mapping (IONM) image is too blurry to understand anything. Figure images should be labelled separately, and caption must be improved.

2.     Number of subjects were just seven for study. Therefore, statistical significance of data is quite not doubtful.

3.      “The authors and conclude that TMS has significantly benefit on improvement of motor impairment, general oncological and comprehensive neurological outcome parameters; moreover, the procedure was safe and well-tolerated by all patients, with no adverse effects neither seizure occurred during or after TMS.”  The statement is confusing.

4.     Discussion section lacks discussion on the results of the study and looks like elaboration of introduction.

Author Response

Thank you very much for your review

  1. Figure 1. The images are blurry. The intraoperative neuromonitoring and mapping (IONM) image is too blurry to understand anything. Figure images should be labelled separately and caption must be improved.

Authors: Thank you for your comment. The purpose of figure 1 was not to provide detailed information about a single case but to demonstrate what is the pipeline of the proof-of-concept study. Nevertheless, we do agree that in the resolution provided the image is not quite clear and therefore the IONM panel has been replaced. Also we have altered the caption of the image providing more details.

  1. Number of subjects were just seven for study. Therefore, statistical significance of data is quite not doubtful.

Authors: Thank you for your comment. The aim of this work is to provide a proof-of-concept study and evaluate the feasibility of this perioperative framework. Our aim was not to provide statistical significant results or compare different perioperative surgical strategies. The above comment has been added to the limitations section of our discussion.

  1. “The authors and conclude that TMS has significantly benefit on improvement of motor impairment, general oncological and comprehensive neurological outcome parameters; moreover, the procedure was safe and well-tolerated by all patients, with no adverse effects neither seizure occurred during or after TMS.”  The statement is confusing.

Authors: Thank you for your comment. This statement was indeed confusing. Therefore, it was reviewed and corrected.

  1. Discussion section lacks discussion on the results of the study and looks like elaboration of introduction.

Authors: Thank you for your comment. The discussion was revised according to the above suggestion. The possible associations and relations of our findings to the mentioned literature were added when appropriate.

Round 2

Reviewer 2 Report

The authors have addressed most of my concerns. There are only several minor format issues  such as:

1. MRI parameters part: b-value = mm2 should be mm2  , unit of FOV should be mm2 .

2. Page5, the volume unit should be mm3

3. Add LR label for Figure 4-6.

Author Response

The authors have addressed most of my concerns. There are only several minor format issues such as:

  1. MRI parameters part: b-value = mm2 should be mm2, unit of FOV should be mm2.

Authors: Thank you very much for your comment. The above has been changed in the manuscript.

  1. Page 5, the volume unit should be mm3

Authors: Thank you very much for your comment. The volume of the tumors is calculated in cm3.

  1. Add LR label for Figure 4-6.

Authors: Thank you very much for your comment. Left and right labels added accordingly. For figure 6, we have separated the images in A and B.

Reviewer 3 Report

Thank you for this improved manuscript with clarifications

Author Response

Thank you very much for your response and your comments.

Reviewer 4 Report

This revised version of the manuscript responses the reviewers' comments satisfactorily.Based on this, I would recommend for acceptance of this article in current form.  

Author Response

(The authors gave the same response as above.)
